# Analysis of case fatality rate of SARS-CoV-2 infection in the Spanish Autonomous Communities between March and May 2020

Martín-Sánchez V.[1,2], Calderón-Montero A. [3]*, Barquilla-García A.[4], Vitelli-Storelli F.[5], Segura-Fragoso A.[6], Olmo-Quintana V. [7], Serrano-Cumplido A.[8], on behalf of the COVID-19 Group of the Spanish Society of Primary Care Physicians (SEMERGEN)

**1** Research Group in Gene-Environment Interactions and Health, Institute of Biomedicine (IBIOMED), University of León, León, Spain, **2** Consortium for Biomedical Research in Epidemiology and Public Health (CIBERESP), Madrid, Spain, **3** Doctor Pedro Laín Entralgo Health Center, Madrid Health Service (SERMAS), Alcorcón, Madrid, Spain, **4** Trujillo Health Center, Extremadura Health Service, Cáceres, Spain, **5** Research Group in Gene-Environment Interactions and Health, Institute of Biomedicine (IBIOMED), University of León, León, Spain, **6** Health Science Department, Castilla La Mancha University, Talavera de la Reina, Toledo, Spain, **7** Management Pharmacy Service Primary Health Care, Vicepresident of Ethical Committee Research with Medicine Hospital Dr. Negrín Gran Canaria Las Palmas (CEI/CEIm) Canary Health Service, Las Palmas, Spain, **8** Family and Community Medicine, Getxo, Bizkaia, Spain

* acalderonmontero@gmail.com

**Data Availability Statement:** All relevant data are within the manuscript and its Supporting Information files.

## Abstract

### Objective

The Spanish health system is made up of seventeen regional health systems. Through the official reporting systems, some inconsistencies and differences in case fatality rates between Autonomous Communities (CC.AA.) have been observed. Therefore the objective of this paper is to compare COVID-19 case fatality rates across the Spanish CC.AA.

### Material and methods

Observational descriptive study. The COVID-19 case fatality rate (CFR) was estimated according to the official records (CFR-PCR+), the daily mortality monitory system (MoMo) record (CFR-Mo), and the seroprevalence study ENE-COVID-19 (Estudio Nacional de sero Epidemiologia Covid-19) according to sex, age group and CC.AA. between March and June 2020. The main objective is to detect whether there are any differences in CFR between Spanish Regions using two different register systems, i. e., the official register of the Ministry of Health and the MoMo.

### Results

Overall, the CFR-Mo was higher than the CFR-PCR+, 1.59% vs 0.98%. The differences in case fatality rate between both methods were significantly higher in Castilla La Mancha, Castilla y León, Cataluña, and Madrid. The difference between both methods was higher in persons over 74 years of age (CFR-PCR+ 7.5% vs 13.0% for the CFR-Mo) but without statistical significance. There was no correlation of the estimated prevalence of infection with CFR-PCR+, but there was with CFR-Mo ($R^2 = 0.33$). Andalucía presented a SCFR below 1

**Funding:** The author(s) received no specific founding for this work

**Competing interests:** The authors have declared that no competing interests exist.

with both methods, and Asturias had a SCFR higher than 1. Cataluña and Castilla La Mancha presented a SCFR greater than 1 in any scenario of SARS-CoV-2 infection calculated with SCFR-Mo.

## Conclusions

The PCR+ case fatality rate underestimates the case fatality rate of the SARS-CoV- 2 virus pandemic. It is therefore preferable to consider the MoMo case fatality rate. Significant differences have been observed in the information and registration systems and in the severity of the pandemic between the Spanish CC.AA. Although the infection prevalence correlates with case fatality rate, other factors such as age, comorbidities, and the policies adopted to address the pandemic can explain the differences observed between CC.AA.

## Introduction

The spread of the pandemic of the severe acute respiratory syndrome coronavirus 2 (SARS-CoV- 2) has affected, up to October 2020, more than 37 million people in 188 countries, and has caused more than one million deaths worldwide [1]. The distribution of the infection is being very heterogeneous, not only across countries but also within nations. The reasons for this variability have not been explained yet [2,3]. Spain is no exception, and ranks seventh in terms of number of confirmed cases, with more than 800,000 cases, and eighth in terms of deaths, with about 33,000 [4,5]. In Spain, the National Health System (NHS) is characterized by the division of competences between the State Administration and the Autonomous Communities (CC.AA.) with the management of most public health and health care competences transferred to the CC.AA.

The distribution of the pandemic is heterogeneous, and so is its severity [2]. The mortality rates and case fatality rates differ widely due not only to an unequal spread of the infection and the different sociodemographic characteristics of populations, but also to the varying criteria for the identification of deceases, cases and infections [6–8]. While this diagnosis variability has been minimized as a result of having centralized the management of the first wave of the pandemic in the Ministry of Health, differences have been observed in the distribution and severity of the infection across CC.AA. [9].

In order to estimate the case fatality rate, it is necessary to know the prevalence of infection in the population. This prevalence has been estimated in Spain with the publication of the seroprevalence study ENE-COVID-19 [10]. This is an epidemiological study stratified by age, sex, and CC.AA., conducted between late April and early May in non-institutionalized population, which estimated the infection prevalence through IgG antibody detection in a sample of more than 60,000 people. Using these data, the case fatality rate in Spain, taking into account only the cases detected with RT-PCR (Reverse Transcription-Polymerase Chain Reaction), was 9.6 deaths per 1,000 infected, ranging from 1/1,000 in the autonomous city of Melilla and 26.6/1,000 in La Rioja [11].

However, the register of deaths only by detection of PCR+ probably underestimates the true scale of the pandemic's case fatality rate. Thus, the daily mortality monitoring system (MoMo), which analyzes observed and estimated mortality according to a historical series of more than 10 years, registers an excess mortality in Spain during the period March to May 2020 above 56%, with about 45,000 deaths more than expected [12]. Therefore there would be

a difference of more than 17,000 deaths between the official records and the estimations by excess deaths. Nevertheless, this excess mortality may include the direct and indirect causes of SARS-CoV-2 as well as other unrelated causes which would be reduced by the decrease in deaths due to causes modified by the lockdown of the population [13,14].

With the aim of studying the differences in the SARS-CoV-2 pandemic's severity in Spain, this study analyzes and compares the differences between the case fatality rate estimated from official deaths (CFR-PCR+) and the one estimated from the MoMo register (CFR-Mo) in the Spanish CC.AA., as well as the relationship between infection prevalence and case fatality rate by both models.

## Material and methods

An observational descriptive study has been conducted using the available information on COVID-19 published by the Spanish Ministry of Health and by the Regions, and published by the daily Mortality Monitoring (MoMo) register on excess deaths [12,15]. The data collected come from the Spanish Ministry of Health and are harmonized for all Spanish Regions and are collected within a window of one week, in order to minimize the possible bias in the report of cases and deaths by the Regions. The MoMo data were collected from de National Register of Deaths published by Carlos III Health Institute, and include a daily monitoritation of all death causes. Since 2020, this data collected all death causes from 3,929 computerized civil registries, representing 92% of the Spanish population. Estimates of expected mortality are made using restrictive models of historical means based on the mortality observed from January 1, 2008 to one year prior to the current date, from the National Institute of Statistics. We have carried out analyses stratified by sex, age groups (<65 years, 65–74 years, and older than 74), and CC. AA. Since the data of the different epidemiological series of the COVID-19 pandemic are provided by decades, it was necessary to assume that both deaths and infections in the ranges 60–69 years and 70–79 years were evenly distributed, re-allocating them proportionally to the ranges <65 years, 65–74 years, and older than 74 years.

### Prevalence of SARS-CoV-2 infection

From the data on prevalence of positive IgG antibody tests of the ENE-COVID-19 study (10), the infection prevalence was estimated considering the validity of the test used in a scenario of 82.1% sensitivity and 100% specificity for the detection of SARS-CoV-2-specific IgG antibodies. The infection prevalence was calculated by sex and age groups for each CC.AA. The results on prevalence are expressed in percentage with the corresponding 95% confidence intervals.

### Estimated number of SARS-CoV-2 infections

It was calculated multiplying the estimated infection prevalence by the corresponding population for each level of analysis (sex, age group, and CC.AA.) obtained from the database of the Spanish National Statistics Institute (INE) on 1 January 2020 [16].

### Deaths

In the Spanish register system, every person who dies with a positive diagnostic result for SARS-CoV-2 was considered a death attributable to covid19, and that is how it has been considered for the analysis. The number of deaths for the calculation of CFR-PCR+ was obtained from the information provided by the Ministry of Health during the week of 11 to 16 May 2020. The number of deaths by sex and age ranges of the different CC.AA was obtained from an internet search on the different web pages of Public Health and of the Health Departments

of the CC.AA. The data used in this analysis were adjusted to the epidemiological situation of each CC.AA in the week of 11 to 16 May 2020. The date of collection of data on cases and deaths was chosen on purpose to make it coincide with the completion of ENE-Covid19 (data collection between 2020/04/27 and 2020/05/11), and therefore, with the highest infection incidence known at that time.

From the daily reports of the MoMo system, the excess deaths from the state of alert to 6 July 2020 were obtained for each CC.AA and according to sex and age group, selecting those dates which presented a higher number of excess deaths in each case [12].

## Case fatality rate

The CFR-PCR+ was calculated by dividing the number of deaths announced by the Ministry of Health by the total number of people infected for each level of analysis (CC.AA, sex and age group). On the other hand, the CFR-Mo was obtained by dividing the excess deaths by the total number of people infected for each level of analysis (CC.AA, sex and age group). In both cases the case fatality rate was expressed as a ratio per 100 estimated infections in the population, and was calculated both for the mean and for the upper and lower ranges of the 95% CI of the infection prevalence reported by the ENE-COVID-19 for each level of analysis.

## Standardized case fatality ratio

By applying the case fatality rate for each age group obtained for the total of Spain, the expected deaths were estimated for each CC.AA. Thus, the standardized case fatality ratio was obtained, together with its corresponding 95% CI by the two methods (CFR-PCR+ and CFR-Mo) with the program EPIDAT [17].

## Correlation between prevalence of infection and case fatality rate

In order to estimate the correlation between the CFR-PCR+ and the CFR-Mo with the prevalence of infection, linear regression techniques were used for an independent variable adjusting for population and stratifying by CC.AA and age groups (program STATA) [18]. Spearman's Rho was also used to analyze the correlation of the estimated infection prevalence with the case fatality rate of the CC.AA and for each age group.

## Ethical considerations

This study was conducted following the protocols and guidelines of good clinical practice according to current legislation.

## Results

The estimated prevalence of SARS-CoV-2 infection was 5.9% (5.5%-6.4%), with an estimated total number of infections of 2,798,444 (Table 1). The official number of deaths from COVID-19 stood at 26,736, whereas the excess deaths estimated by MoMo rose to 44,676 (Table 2). Based on these data, the estimated CFR-PCR+ was 0.98% (0.91–1.04) and the CFR-Mo was 1.59% (1.48–1.69) (Fig 1). There were relevant differences in case fatality rates between CC.AA by both methods, ranging from the minimum of 0.26% and 0.25% of Islas Canarias and the maximum of 2.82% and 2.44% of La Rioja (CFR-PCR+ and CFR-Mo, respectively). The CFR-Mo was significantly higher to CFR-PCR+ in Castilla La Mancha, Castilla y León, Cataluña, and Madrid. However, it was lower, though not significantly, in Canarias, Galicia, Islas Baleares, and La Rioja (Fig 1).

**Table 1. Population distribution and prevalences of infection by Autonomous Communities (CC.AA).**

| CCAA | Global | | | | Hombres | | | | Mujeres | | | | Menos de 65 años | | | | 65 a 74 años | | | | 75 años y más | | | |
|---|---|---|---|---|---|---|---|---|---|---|---|---|---|---|---|---|---|---|---|---|---|---|---|---|
| | P | Pr | Ls | Li | P | Pr | Ls | Li | P | Pr | Ls | Li | P | Pr | Ls | Li | P | Pr | Ls | Li | P | Pr | Ls | Li |
| AND | 8,460,261 | 3.1 | 3.8 | 2.6 | 4,168,872 | 3.2 | 3.9 | 2.6 | 4,291,389 | 3.1 | 3.9 | 2.5 | 6,989,583 | 2.9 | 5.3 | 1.8 | 772,301 | 4.0 | 6.8 | 2.4 | 698,377 | 3.3 | 5.3 | 1.3 |
| ARA | 1,328,753 | 5.8 | 7.5 | 4.5 | 655,734 | 6.0 | 8.0 | 4.5 | 673,019 | 5.7 | 7.8 | 4.0 | 1,040,102 | 5.4 | 12.3 | 2.5 | 136,578 | 7.5 | 15.9 | 3.4 | 152,073 | 5.9 | 16.9 | 2.0 |
| AST | 1,018,706 | 2.1 | 2.9 | 1.5 | 486,031 | 2.2 | 2.9 | 1.3 | 532,675 | 2.0 | 2.9 | 1.3 | 752,144 | 2.2 | 9.1 | 0.6 | 133,018 | 1.4 | 8.1 | 0.2 | 133,544 | 1.8 | 4.0 | 0.7 |
| CAN | 582,796 | 3.8 | 5.9 | 2.4 | 282,517 | 2.7 | 4.6 | 1.7 | 300,279 | 4.7 | 8.0 | 2.8 | 453,377 | 3.7 | 11.1 | 1.3 | 65,995 | 3.2 | 11.2 | 1.1 | 63,424 | 5.0 | 20.8 | 1.2 |
| CAT | 7,778,362 | 6.8 | 8.1 | 5.8 | 3,825,977 | 7.1 | 8.7 | 5.8 | 3,952,385 | 6.7 | 8.0 | 5.5 | 6,310,129 | 6.5 | 10.4 | 4.0 | 742,909 | 9.8 | 15.2 | 6.2 | 725,324 | 8.1 | 15.0 | 4.2 |
| CEU | 83,842 | 1.3 | 2.9 | 0.6 | 42,370 | 1.5 | 4.1 | 0.6 | 41,472 | 1.2 | 2.6 | 0.5 | 73,698 | 1.7 | 7.8 | 0.4 | 5,636 | 0.0 | 0.0 | 0.0 | 4,508 | 0.0 | 0.0 | 0.0 |
| CLM | 2,044,408 | 12.7 | 14.6 | 11.0 | 1,023,399 | 11.7 | 13.7 | 9.9 | 1,021,009 | 13.8 | 16.2 | 11.7 | 1,654,187 | 12.4 | 16.9 | 8.1 | 181,715 | 21.1 | 28.6 | 13.7 | 208,506 | 11.0 | 21.1 | 7.4 |
| CVA | 5,054,796 | 2.9 | 3.8 | 2.2 | 2,490,903 | 3.3 | 4.4 | 2.5 | 2,563,893 | 2.6 | 3.5 | 1.9 | 4,073,243 | 2.8 | 6.5 | 1.2 | 510,863 | 4.4 | 9.8 | 2.4 | 470,690 | 3.7 | 10.6 | 0.8 |
| CYL | 2,393,285 | 8.5 | 9.7 | 7.4 | 1,178,111 | 7.8 | 9.1 | 6.6 | 1,215,174 | 9.2 | 10.6 | 7.9 | 1,779,687 | 8.5 | 12.1 | 5.5 | 277,294 | 9.5 | 13.1 | 6.1 | 336,304 | 8.5 | 15.2 | 4.7 |
| EXT | 1,063,575 | 3.5 | 4.8 | 2.6 | 526,101 | 3.3 | 4.8 | 2.2 | 537,474 | 3.7 | 5.3 | 2.6 | 840,349 | 3.2 | 7.1 | 1.1 | 105,042 | 7.2 | 13.4 | 2.8 | 118,184 | 6.5 | 17.9 | 2.4 |
| GAL | 2,700,269 | 2.5 | 3.1 | 2.0 | 1,299,371 | 2.5 | 3.4 | 1.8 | 1,400,898 | 2.5 | 3.5 | 1.9 | 2,012,584 | 2.4 | 5.3 | 0.9 | 322,962 | 2.8 | 7.5 | 1.7 | 364,723 | 4.1 | 8.8 | 1.3 |
| IBA | 1,171,003 | 2.9 | 4.2 | 2.0 | 584,094 | 2.5 | 4.0 | 1.4 | 586,909 | 3.4 | 5.7 | 2.1 | 987,872 | 2.7 | 10.0 | 0.8 | 99,393 | 3.8 | 13.2 | 1.1 | 83,738 | 4.0 | 21.8 | 0.7 |
| ICA | 2,174,474 | 2.0 | 2.9 | 1.3 | 1,075,496 | 2.2 | 3.4 | 1.5 | 1,098,978 | 1.8 | 2.9 | 0.9 | 1,823,421 | 2.2 | 6.6 | 0.8 | 193,834 | 1.4 | 4.1 | 0.5 | 157,219 | 2.4 | 5.2 | 0.1 |
| LRI | 319,653 | 3.9 | 5.3 | 2.8 | 157,699 | 4.0 | 6.1 | 2.6 | 161,954 | 3.7 | 6.0 | 2.2 | 252,348 | 4.1 | 11.0 | 1.5 | 32,442 | 3.4 | 14.4 | 1.7 | 34,863 | 1.3 | 15.9 | 0.4 |
| MAD | 6,778,382 | 13.3 | 15.3 | 11.6 | 3,243,153 | 13.3 | 15.7 | 11.1 | 3,535,229 | 13.3 | 15.7 | 11.3 | 5,569,606 | 12.7 | 19.5 | 8.1 | 613,064 | 18.5 | 25.3 | 11.0 | 595,712 | 10.4 | 27.6 | 7.7 |
| MEL | 87,076 | 2.2 | 3.5 | 1.4 | 44,173 | 1.9 | 5.0 | 0.7 | 42,903 | 2.5 | 4.6 | 1.4 | 77,870 | 2.2 | 12.0 | 0.5 | 5,164 | 5.0 | 23.5 | 0.9 | 4,042 | 0.0 | 0.0 | 0.0 |
| MUR | 1,510,951 | 1.7 | 2.8 | 0.9 | 756,619 | 1.5 | 3.3 | 0.7 | 754,332 | 1.8 | 3.9 | 0.8 | 1,273,017 | 1.5 | 7.1 | 0.6 | 123,639 | 9.2 | 17.2 | 2.5 | 114,295 | 0.0 | 0.0 | 0.0 |
| NAV | 660,887 | 6.8 | 9.1 | 5.1 | 327,073 | 7.1 | 10.3 | 5.0 | 333,814 | 6.5 | 9.2 | 4.5 | 530,226 | 6.1 | 13.8 | 2.9 | 65,113 | 9.2 | 17.2 | 2.5 | 65,548 | 11.7 | 26.3 | 2.9 |
| PVA | 2,219,777 | 4.7 | 6.1 | 3.7 | 1,079,024 | 4.4 | 5.9 | 3.2 | 1,140,753 | 5.1 | 7.0 | 3.8 | 1,720,349 | 4.5 | 10.1 | 2.0 | 248,729 | 5.9 | 13.0 | 3.1 | 250,699 | 6.0 | 15.1 | 1.8 |
| ESP | 47,431,256 | 5.9 | 6.4 | 5.5 | 23,246,717 | 5.9 | 6.4 | 5.4 | 24,184,539 | 6.0 | 6.5 | 5.5 | 38,213,792 | 5.8 | 7.0 | 4.6 | 4,635,691 | 7.7 | 9.2 | 6.4 | 4,581,773 | 6.2 | 9.3 | 4.8 |

P = population; Pr = prevalence of infection per 100 inhabitants; UR and LR = Upper and lower ranges of the 95% confidence interval.

Table 2. Death figures and case fatality rates according to official data and calculated according to excess mortality, by Autonomous Communities (CC.AA.) and gender.

| CC.AA. | Global | | | | | | | | Men | | | | | | | | Women | | | | | | | |
|---|---|---|---|---|---|---|---|---|---|---|---|---|---|---|---|---|---|---|---|---|---|---|---|---|
| | Official Data | | | | MoMo Mortality | | | | Official Data | | | | MoMo Mortality | | | | Official Data | | | | MoMo Mortality | | | |
| | D | CFR | UR | LR | D | CFR | UR | LR | D | CFR | UR | LR | D | CFR | UR | LR | D | CFR | UR | LR | D | CFR | UR | LR |
| AND | 1358 | 052 | 0.62 | 0.43 | 1756 | 068 | 0.80 | 0.55 | 743 | 056 | 0.69 | 0.46 | 663 | 050 | 0.61 | 0.41 | 583 | 044 | 0.55 | 0.35 | 1080 | 082 | 1.02 | 0.65 |
| ARA | 859 | 112 | 144 | 086 | 945 | 123 | 159 | 094 | 455 | 115 | 155 | 087 | 430 | 109 | 146 | 082 | 404 | 106 | 150 | 077 | 458 | 120 | 170 | 087 |
| AST | 317 | 147 | 203 | 106 | 423 | 196 | 271 | 141 | 171 | 157 | 271 | 090 | 148 | 136 | 235 | 078 | 146 | 137 | 211 | 093 | 233 | 218 | 337 | 148 |
| CAN | 209 | 095 | 152 | 061 | 212 | 096 | 154 | 062 | 98 | 128 | 210 | 075 | 142 | 185 | 305 | 109 | 111 | 078 | 131 | 046 | 128 | 090 | 151 | 053 |
| CAT | 5956 | 112 | 133 | 094 | 11690 | 220 | 260 | 185 | 2877 | 106 | 130 | 086 | 5648 | 209 | 256 | 169 | 3079 | 116 | 141 | 097 | 6291 | 237 | 287 | 199 |
| CEU | 4 | 037 | 081 | 016 | 11 | 101 | 223 | 045 | 2 | 031 | 080 | 011 | 9 | 139 | 360 | 051 | 2 | 041 | 102 | 019 | 2 | 041 | 102 | 019 |
| CLM | 2898 | 111 | 129 | 097 | 5314 | 204 | 237 | 178 | 1507 | 126 | 149 | 108 | 2640 | 221 | 260 | 189 | 1391 | 099 | 117 | 084 | 2557 | 182 | 215 | 155 |
| CVA | 1370 | 092 | 121 | 072 | 1740 | 117 | 154 | 091 | 761 | 093 | 123 | 070 | 902 | 110 | 146 | 083 | 609 | 092 | 126 | 067 | 872 | 131 | 180 | 096 |
| CYL | 1953 | 096 | 110 | 084 | 3615 | 178 | 203 | 156 | 1113 | 121 | 143 | 104 | 1886 | 206 | 242 | 176 | 840 | 075 | 088 | 065 | 1768 | 158 | 184 | 137 |
| EXT | 500 | 133 | 181 | 097 | 702 | 187 | 254 | 137 | 291 | 168 | 247 | 114 | 274 | 158 | 232 | 103 | 209 | 106 | 150 | 073 | 372 | 189 | 267 | 130 |
| GAL | 607 | 091 | 112 | 073 | 582 | 087 | 108 | 070 | 381 | 118 | 166 | 086 | 242 | 075 | 105 | 054 | 226 | 065 | 086 | 046 | 268 | 077 | 101 | 054 |
| IBA | 218 | 063 | 093 | 044 | 210 | 061 | 089 | 042 | 135 | 093 | 163 | 058 | 162 | 112 | 196 | 069 | 83 | 041 | 067 | 025 | 86 | 043 | 069 | 026 |
| ICA | 153 | 035 | 054 | 024 | 146 | 033 | 052 | 023 | 86 | 036 | 052 | 023 | 68 | 028 | 041 | 018 | 67 | 034 | 065 | 021 | 29 | 015 | 028 | 009 |
| LRI | 351 | 282 | 388 | 207 | 304 | 244 | 336 | 179 | 192 | 304 | 469 | 199 | 169 | 267 | 413 | 175 | 159 | 269 | 438 | 163 | 142 | 240 | 391 | 146 |
| MAD | 8521 | 094 | 109 | 082 | 14308 | 158 | 183 | 138 | 4431 | 103 | 123 | 087 | 7172 | 166 | 200 | 141 | 4090 | 087 | 102 | 074 | 6407 | 136 | 160 | 116 |
| MEL | 2 | 010 | 016 | 006 | ND | ND | ND | ND | 2 | 024 | 064 | 009 | ND | ND | ND | ND | 0 | 000 | 000 | 000 | ND | ND | ND | ND |
| MUR | 145 | 058 | 102 | 034 | 248 | 099 | 174 | 058 | 81 | 070 | 151 | 032 | 120 | 103 | 224 | 048 | 64 | 048 | 103 | 022 | 136 | 102 | 218 | 046 |
| NAV | 511 | 113 | 153 | 085 | 683 | 151 | 204 | 114 | 273 | 118 | 169 | 081 | 302 | 131 | 186 | 090 | 238 | 110 | 159 | 078 | 357 | 165 | 239 | 116 |
| PVA | 1442 | 138 | 178 | 106 | 1607 | 154 | 198 | 118 | 723 | 154 | 210 | 114 | 716 | 152 | 208 | 113 | 719 | 124 | 167 | 091 | 718 | 124 | 167 | 090 |
| SPAIN | 27374 | 098 | 104 | 091 | 44546 | 159 | 169 | 148 | 14322 | 105 | 114 | 097 | 22162 | 162 | 176 | 150 | 13020 | 090 | 097 | 083 | 22554 | 155 | 168 | 144 |

D = number of deaths; CFR = case fatality rate per 100 infections; UR and LR = Upper and lower ranges of the 95% confidence interval. ND = data not reported.

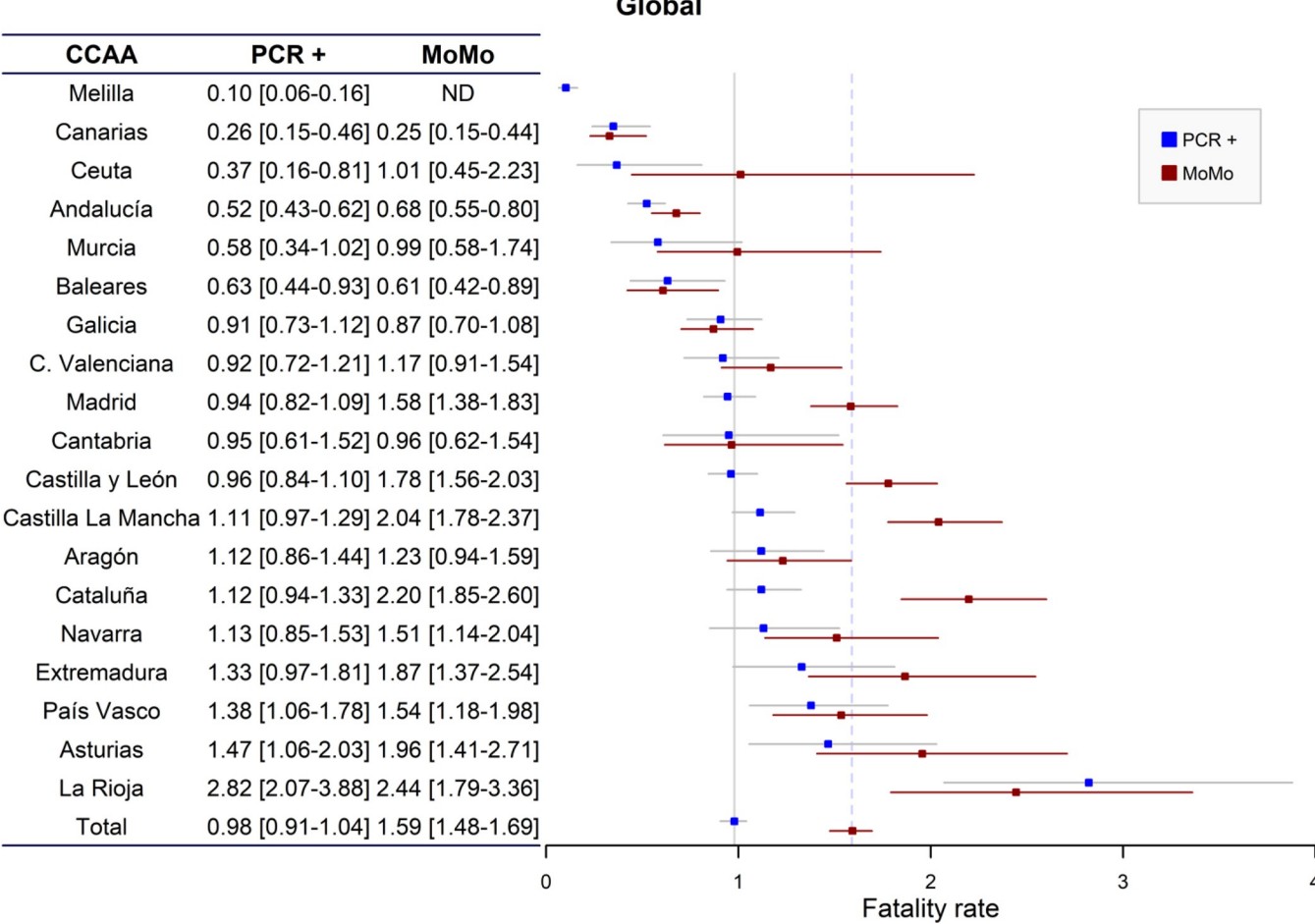

**Fig 1. Case fatality rate with their ranges according to prevalence of infection calculated by official number of deaths (PCR+) and by excess mortality (MoMo) in the different Autonomous Communities (CC.AA).**

## Gender

No differences were observed between men and women in the estimated prevalences of SARS-CoV-2 infection (5.9%; 5.4–6.4 vs 6.0%; 5.5–6.5). Nevertheless, the CFR-PCR+ was significantly higher in men (10.5%; 0.97–1.14) as compared with women (0.90%; 0.83–0.97), whereas the CFR- Mo did not show significant differences (1.62%; 1.50–1.70 in men and 1.55%; 1.44–1.68 in women). Overall, we can assume that the distribution of the case fatality rate by sex in the different CC.AA reproduces what has been seen in the global analysis (Table 2).

## Age

There were no significant differences in the estimated prevalences of SARS-CoV-2 infection when analyzed by age groups (5.8%, 7.7%, and 6.2% for those younger than 65 years, 65–74 years, and older than 74 years, respectively) (Table 1). The estimated infections and deaths are presented in Table 3. The analysis by age groups showed that the case fatality rate in those under the age of 65 was approximately 1/1000 infections, regardless of the method used (0.09% [0.08–0.12] and 0.10% [0.09–0.13]; CFR-PCR+ and CFR-Mo, respectively); 1/100 in those between 65 and 74 years old, but with significant differences according to the method

**Table 3. Death figures and case fatality rates according to official data calculated according to excess mortality, by Autonomous Communities (CC.AA.) and age groups.**

| CC.AA | Under age 65 | | | | | | | | Age 65 to 74 | | | | | | | | Age 75 and older | | | | | | | |
|---|---|---|---|---|---|---|---|---|---|---|---|---|---|---|---|---|---|---|---|---|---|---|---|---|
| | Official Data | | | | MoMo Mortality | | | | Official Data | | | | MoMo Mortality | | | | Official Data | | | | MoMo Mortality | | | |
| | D | CFR | UR | LR | D | CFR | UR | LR | D | CFR | UR | LR | D | CFR | UR | LR | D | CFR | UR | LR | D | CFR | UR | LR |
| AND | 166 | **008** | 0.13 | 0.04 | 206 | **010** | 0.17 | 0.06 | 204 | **066** | 112 | 039 | 119 | **038** | 065 | 023 | 911 | **40** | 101 | 25 | 1371 | **59** | 151 | 37 |
| ARA | 73 | **013** | 028 | 006 | 69 | **012** | 027 | 005 | 98 | **095** | 210 | 045 | 82 | **080** | 176 | 038 | 688 | **77** | 226 | 27 | 813 | **91** | 267 | 32 |
| AST | 17 | **010** | 038 | 002 | 16 | **009** | 036 | 002 | 38 | **202** | 1,212 | 035 | 42 | **223** | 1,339 | 039 | 262 | **111** | 277 | 49 | 405 | **171** | 429 | 76 |
| CAN | 10 | **006** | 017 | 002 | 0 | **000** | 000 | 000 | 24 | **114** | 343 | 032 | 12 | **057** | 171 | 016 | 174 | **55** | 233 | 13 | 181 | **58** | 242 | 14 |
| CAT | 454 | **011** | 018 | 007 | 576 | **014** | 023 | 009 | 635 | **087** | 137 | 056 | 1304 | **179** | 281 | 115 | 4886 | **83** | 159 | 45 | 9955 | **169** | 323 | 92 |
| CEU | 0 | **000** | 000 | 000 | 1 | **008** | 038 | 002 | 0 | **NA** | NA | NA | 3 | **NA** | NA | NA | 4 | **NA** | NA | NA | 7 | **NA** | NA | NA |
| CLM | 278 | **014** | 021 | 010 | 408 | **020** | 030 | 015 | 140 | **037** | 056 | 027 | 690 | **180** | 278 | 133 | 2478 | **108** | 160 | 56 | 4272 | **187** | 276 | 97 |
| CVA | 130 | **011** | 027 | 005 | 96 | **008** | 020 | 004 | 237 | **106** | 197 | 047 | 248 | **111** | 206 | 050 | 1019 | **59** | 262 | 20 | 1394 | **81** | 359 | 28 |
| CYL | 186 | **012** | 019 | 009 | 131 | **009** | 013 | 006 | 117 | **044** | 069 | 032 | 311 | **117** | 183 | 086 | 1633 | **57** | 103 | 32 | 3166 | **111** | 200 | 62 |
| EXT | 45 | **017** | 050 | 008 | 0 | **000** | 000 | 000 | 30 | **040** | 101 | 021 | 58 | **077** | 195 | 041 | 425 | **55** | 153 | 20 | 652 | **85** | 234 | 31 |
| GAL | ND | **ND** | ND | ND | 0 | **000** | 000 | 000 | ND | **ND** | ND | ND | 16 | **018** | 030 | 007 | ND | **ND** | ND | ND | 460 | **31** | 97 | 14 |
| IBA | 27 | **010** | 033 | 003 | 0 | **000** | 000 | 000 | 35 | **093** | 332 | 027 | 56 | **149** | 531 | 043 | 162 | **48** | 273 | 09 | 122 | **36** | 206 | 07 |
| ICA | 17 | **004** | 011 | 001 | 39 | **010** | 026 | 003 | 33 | **120** | 361 | 041 | 56 | **204** | 613 | 070 | 103 | **28** | 556 | 13 | 52 | **14** | 281 | 06 |
| LRI | 41 | **039** | 106 | 015 | 16 | **015** | 041 | 006 | 28 | **252** | 523 | 060 | 18 | **162** | 336 | 039 | 298 | **659** | 2,417 | 54 | 214 | **473** | 1,000 | 39 |
| MAD | 593 | **008** | 013 | 005 | 909 | **013** | 020 | 008 | 1288 | **114** | 192 | 083 | 1842 | **162** | 274 | 119 | 6640 | **107** | 145 | 40 | 11448 | **185** | 251 | 70 |
| MEL | 0 | **000** | 000 | 000 | ND | **ND** | ND | ND | 0 | **000** | 000 | 000 | ND | **ND** | ND | ND | 2 | **NA** | NA | NA | ND | **ND** | ND | ND |
| MUR | 11 | **006** | 015 | 001 | 0 | **000** | 000 | 000 | 13 | **NA** | NA | NA | 67 | **NA** | NA | NA | 121 | **NA** | NA | NA | 205 | **NA** | NA | NA |
| NAV | 24 | **007** | 015 | 003 | 13 | **004** | 008 | 002 | 63 | **105** | 391 | 056 | 72 | **120** | 447 | 064 | 424 | **55** | 219 | 25 | 590 | **77** | 305 | 34 |
| PVA | 159 | **021** | 046 | 009 | 110 | **014** | 032 | 006 | 254 | **173** | 333 | 079 | 152 | **104** | 199 | 047 | 1172 | **78** | 264 | 31 | 1314 | **87** | 296 | 35 |
| SPAIN | 2097 | **009** | 012 | 008 | 2308 | **010** | 013 | 009 | 3237 | **091** | 110 | 076 | 5132 | **144** | 174 | 120 | 21402 | **75** | 97 | 50 | 37236 | **130** | 168 | 87 |

D = number of deaths; CFR = case fatality rate per 100 infections; UR and LR = Upper and lower ranges of the 95% confidence interval. NA = not applicable; ND = data not reported.

used (0.91% [0.76–1.10] and 1.44% [1.20–1.74]; CFR-PCR+ and CFR-Mo, respectively); and 1/10 in those over the age of 74, but with non-significant differences according to the method used (7.5% [5.0–9.7] and 13.0% [8.7–16.8]; CFR-PCR+ and CFR-Mo, respectively).

The differences observed in the estimated case fatality rate by the two methods (CFR-PCR+ or CFR-Mo) increase with age, so that, though they are not significant in those under 65 (0.09 [0.08–0.12] vs 0.10 [0.09–0.13]), they are significant in those between 65 and 74 (0.91 [0.76–1.10] vs 1.44 [1.20–1.74]), and the CFR-Mo nearly doubles the CFR-PCR+ in those older than 74 (7.5 [5.0–9.7] vs 13.0 [8.7–16.8]). The same can be seen in the CC.AA (Table 3 and Fig 2).

## Standardized case fatality ratio

In the analysis of the standardized case fatality ratio (SCFR) (Fig 3), only two CC.AA present similar results with both methods of case fatality rate estimation. Andalucía was the only CC.AA where the number of deaths observed with both methods is significantly lower than expected (SCFR by PCR+ 0.54 [0.66–0.71] and SCFR by MoMo 0.46 [0.61–0.56]). On the other hand, Asturias was the only Region where the number of deaths observed is significantly higher than expected (SCFR-PCR+ 1.41 [1.10–1.97] and SCFR-Mo 1.31 [1.05–1.77]). In the case of SCFR-Mo, Cataluña 1.29 [1.57–1.23] and Castilla La Mancha 1.43 [1.72–1.10] presented values higher than 1 both in the upper and lower ranges of estimated SARS-CoV-2 infection.

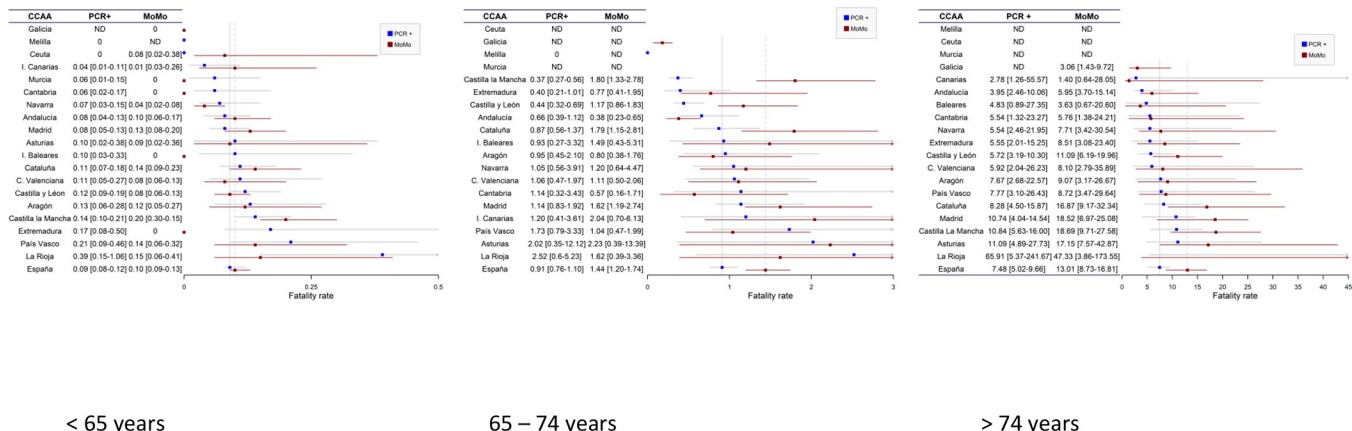

|  | < 65 years | 65 – 74 years | > 74 years |

**Fig 2. Case fatality rates with ranges according to prevalence of infection and age groups calculated by official number of deaths (PCR+) and by excess mortality (MoMo) in the different Autonomous Communities (CC.AA).**

## Standardized Mortality Ratio

| CCAA | PCR+ | MoMo |
|---|---|---|
| Galicia | ND | 0.22 [0.23-0.34] |
| I. Canarias | 0.42 [0.38-2.17] | 0.26 [0.24-1.45] |
| Andalucía | 0.54 [0.71-0.66] | 0.46 [0.61-0.56] |
| I. Baleares | 0.67 [0.29-1.80] | 0.34 [0.15-0.91] |
| Extremadura | 0.70 [0.59-1.00] | 0.63 [0.52-0.87] |
| Castilla y León | 0.72 [0.88-0.66] | 0.85 [1.05-0.76] |
| Cantrabria | 0.72 [0.39-1.50] | 0.42 [0.23-0.87] |
| Navarra | 0.73 [0.72-1.43] | 0.60 [0.60-1.18] |
| C. Valenciana | 0.81 [0.64-1.57] | 0.65 [0.51-1.24] |
| Aragón | 0.98 [0.79-1.45] | 0.70 [0.56-1.01] |
| Cataluña | 1.02 [1.24-0.99] | 1.29 [1.57-1.23] |
| País Vasco | 1.11 [0.99-1.81] | 0.70 [0.62-1.12] |
| Castilla la Mancha | 1.21 [1.45-0.95] | 1.43 [1.72-1.10] |
| Madrid | 1.26 [1.18-0.94] | 1.35 [1.25-0.99] |
| Asturias | 1.41 [1.10-1.97] | 1.31 [1.05-1.77] |
| La Rioja | 6.39 [1.55-10.57] | 2.86 [0.66-4.69] |

**Fig 3. Distribution of the standardized case fatality ratio (SCFR) and ranges according to prevalence of infection calculated from de ENE-COVID-19 study and adjusted for age group in the different Autonomous Communities (CC.AA).**

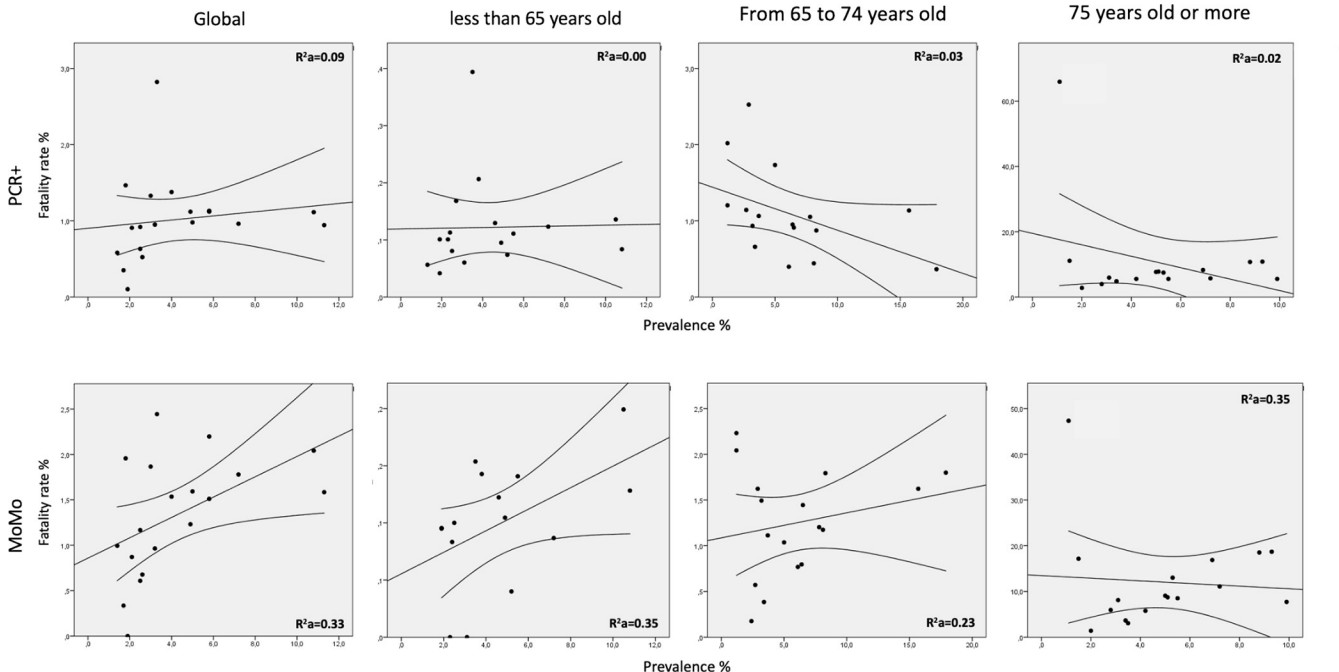

**Fig 4. Correlation between prevalence of infection and case fatality rates by age groups calculated for the official number of deaths (PCR+) and the number of deaths by excess mortality (MoMo).** The straight line represents the linear regression line, and the curves represent the 95% confidence interval of the regression line.

## Correlation between prevalence of infection and case fatality rate

Fig 4 shows the charts with the correlations between the estimated infection prevalence and the case fatality rates of the CC.AA. In the case of the fatality rate estimated from official deaths attributed to COVID-19, the correlations are moderate and even negative in some cases (Table 4), so that the case fatality rate is not explained by the estimated infection prevalence. In the case of the fatality rate estimated from MoMo, the correlation with the estimated infection prevalence is relevant, so that the higher the infection prevalence the higher the case fatality rate (Table 4 and Fig 4). In the case of the correlation between the estimated infection prevalence and the case fatality rate of people over 74 years of age, the correlation adjusting by

**Table 4. Correlation between case fatality rate and prevalence of infection by age groups.**

|  | Global | | Age 18–64 | | Age 65–74 | | Age >74 | |
|---|---|---|---|---|---|---|---|---|
|  | CFR_PCR+ | CFR-Mo | CFR_PCR+ | CFR-Mo | CFR_PCR+ | CFR-Mo | CFR_PCR+ | CFR-Mo |
| Spearman's Rho Coefficient | 0,449 | 0,584 | 0,329 | 0,458 | -0,621 | 0,134 | 0,085 | 0,255 |
| Sig (bilateral) | 0,054 | 0,009 | 0,198 | 0,086 | 0,010 | 0,609 | 0,753 | 0,323 |
| N | 19 | 19 | 17 | 15 | 16 | 17 | 16 | 17 |
| R2 | 00,380 | 02,297 | 00,256 | 03,228 | 01,347 | 04,172 | 01,037 | 00,055 |
| Adjusted R2 | 00,905 | 03,299 | 00,029 | 03,472 | 00,324 | 02,273 | 00,213 | 03,548 |

CFR_PCR+ = Case fatality rate with PCR. CFR-Mo = Case fatality rate with MoMo. N = number of Autonomous Communities included in the analysis. R2 = correlation coefficient.

population indicates that the prevalence of infection explains 35% of the mortality (Spearman's rho without including La Rioja of 0.479; p = 0.071).

## Discussion

A group of Spanish researchers [19] requested some months ago an independent assessment of the Spanish response to the SARS-CoV-2 pandemic, because it has hit Spain very hard and the number infections and deaths cannot be understood in a country that has a well-developed healthcare system. A previous paper which analyzed the case fatality rate with the available official data revealed the heterogeneity in case fatality rates among the different CC.AA, and the need to carry out a study which tried to explained the reason for these differences [20]. The restrictive criteria for the inclusion of COVID-19 deaths in the official statistics, and the little correlation between infection prevalence and case fatality rate suggested the need for a different approach that included in some ways the actual deaths and not only the official ones. In this new paper, the starting hypotheses have been confirmed, and we can affirm that the case fatality rate estimated with the official deaths exclusively is about 50% below the one calculated with the excess mortality of the MoMo system (0.98% vs 1.59%). And indeed there is a positive correlation between case fatality rate and the estimated prevalences of SARS-CoV-2 infection if analyzed with excess mortality. Likewise, large differences can be seen between the case fatality rates estimated from official announced deaths and deaths detected with the excess mortality between the CC.AA, with a close agreement in some cases and a great discrepancy in others.

The main determinant of the case fatality rates calculated in our study is the estimated infection prevalence, based in this case on the ENE-COVID-19 study, and which has shown the large heterogeneity of infection prevalence by CC.AA as seen in other European studies [10]. The varying criteria for establishing the cause of death, the methodological variability, and the socio- demographic characteristics such as predominance of ageing population in Lombardy and in some Spanish CC.AA also determine the case fatality rate of the SARS-CoV-2 infection [20–22]. Pastor-Barriuso et al. [23] have recently published with a similar methodology lower case fatality rate results than our study (0.83% and 1.07% vs 0.98% and 1.59% for CFR-PCR+ and CFR-Mo, respectively). In the study by Pastor-Barriuso the estimated deaths of institutionalized population are excluded, and the calculation of infected population from the ENE-COVID-19 study data does not take into account the validity of the test used. A meta-analysis conducted by Cochrane shows that the validity of the tests has been studied preferably in hospitalized population, and thus there are reasonable doubts about their accuracy when applied in general population [24]. It is also known that, in some asymptomatic patients, the levels of IgG antibodies are not detectable until the second or third week after disease onset, and in addition these decrease very soon, both factors being responsible for the possible false negatives [25,26]. We consider that the estimation of the infected population taking into account the internal validity of the ENE-COVID-19 study (82.1% sensitivity and 100% specificity) offers a more accurate approximation to the actual seroprevalence. Similarly, we consider that the inclusion of institutionalized patients in the calculation of case fatality rates allows a better approximation to the severity of the pandemic. The estimated total number of persons living in nursing homes in Spain is 334,310 (0.7% of the overall population), and the number of deaths in Spanish nursing homes during the first wave of the pandemic, according to the data obtained from the publication of the CC.AA´ records, was more than 27,000; in addition, the number of cases attributed to COVID-19 exceeds 19,000 (about 70% of official deaths) [27]. These figures represent a mortality of 5.7% among institutionalized persons, and if excluded, the actual case fatality rate of COVID-19 is substantially underestimated. Some

authors suggest that more than 20% of community-dwelling people over 65 years of age should be classified as very high risk for COVID-19, and that the higher risk of older population is not exclusive to those living in nursing homes [28]. Moreover, the results about seroprevalence in institutionalized population are very heterogeneous and of a complex methodology. In nursing homes of the United Kingdom, infection prevalences are above 60%, while in other contexts they are about 33% [29–31]. In studies conducted in Spain, the prevalence of infection among doctors and nurses who work in nursing homes doubles the prevalence in those working in primary care (9.5% vs 5.5%) [32]. The analysis of 69 nursing homes (n = 3214 residents) in Barcelona found a prevalence of infection by PCR+ of 23.9% [33]. It follows that, on the whole, a higher seroprevalence should be expected in institutionalized population over 65 as compared with those living in the community. In any case, this would mean an increased denominator of the case fatality rate (infections), so that the case fatality rate would be lower and close to the lower value of the range estimated in our study.

The differences in the case fatality rates calculated with each method are important depending on the age group, and very heterogeneous depending on the CC.AA analyzed. The different objectives of each method explain these differences at least partially. On the one hand, the objective of the official COVID-19 death registration is to assess the impact of the prevention and control measures implemented, and therefore this register is very restrictive as for the case criteria; thus, the case fatality rate obtained is probably underestimated and can be considered as a minimum value of case fatality rate. On the other hand, the objective of the MoMo register is to detect the excess mortality over time, and the cause of death is not known for each particular case. When calculating the CFR-Mo, it is assumed that all excess mortality is attributable to SARS-CoV-2 infection. The SARS-CoV-2 virus has been associated with different respiratory, cardiac, and neurological complications which can be direct causes of mortality [34–37]. On the other hand, there has been a decrease in emergency coronary angiography screening and in stroke care in emergency departments, both in Spain and in other countries [38–40], so MoMo may include causes of death other than SARS-CoV-2 infection. It could therefore be understood that the MoMo register probably leads to an overestimation of case fatality rate, and it can be considered a maximum case fatality rate attributable to SARS-CoV-2.

Although most CC.AA do not present significant differences in case fatality rate according to the method used to estimate it, in Castilla La Mancha, Castilla y León, Cataluña, and Madrid we find very significant differences, with a CFR-PCR+ much lower than the CFR-Mo. This illustrates the problems arising from the structure of the Spanish healthcare system, where the data provided by the CC.AA are not always comparable between the different CC.AA. It also indicates the need for coordination and harmonization from the Ministry of Health, as some Spanish researchers have already requested [41]. The five CC.AA with significant differences in the case fatality rates estimated with each method have very diverse socio-demographic and even socio-economic characteristics, and the only link found was that these CC.AA had the highest SARS-CoV-2 infection prevalences estimated by ENE-COVID-19. It could be expected that with a higher infection prevalence and a larger caseload, the saturation of the system would lead to a higher case fatality rate, particularly among older population. However, the correlation between the estimated infection prevalence and the CFR-PCR+ is small and even negative in some cases. On the contrary, the correlation of the infection prevalence with the CFR-Mo is higher than 33% and relevant for all age groups. Kenyon [42] finds a correlation between the CFR-PCR+ and the estimated infection prevalence in population over 65 years of age, as well as a higher case fatality rate in those CC.AA where the pandemic spread more intensely and quickly, though this study does not estimate the CFR-Mo or analyze people over 75 years of age specifically. We consider, like other studies [43–45], that age and comorbidity

have a more important role than the prevalence of infection, and that those can be the major risk factors in the elderly.

Another key issue in our study emerges from comparing the difference between the CFR-Mo and the CFR-PCR+ by age groups. We observe that this difference increases with age, so that both ways of calculating the case fatality rate give similar results among people under age 65, whereas in those over 74 the CFR-Mo nearly doubles the CFR-PCR+. In population under 65 years of age, with a stronger immune system and a lower comorbidity, individual susceptibility may have been the key mortality factor [46,47]. Additionally, this group has had in general greater accessibility to hospital care, with few out-of-hospital deaths, which explains why there are no differences between both case fatality rates. On the other hand, the older population, and especially those over age 74, with a less efficient immune system and greater comorbidity [48–50], have had a poorer accessibility to hospitals, due to their personal situation, to their disabilities, to their institutionalization, or to the different social and health guidelines, which justifies a higher out-of-hospital mortality and the difference between the CFR-Mo and the CFR- PCR+.

The variability across CC.AA within the same country is a constant feature in the development of the pandemic, without clear reasons that explain it. In people under age 65, the differences between the CFR-PCR+ and the CFR-Mo are minimal between CC.AA. At age 65 and older, and particularly in those over 74, the differences are more evident between CC.AA, but given the wide ranges in the estimated infection prevalences, they do not reach statistical significance.

In the analysis of the standardized case fatality ratio (SCFR), where the effect of the different distribution by age groups has been controlled, the case of Andalucía stands out, which consistently shows an overall case fatality rate lower than the national average. In contrast, La Rioja shows an increased case fatality rate both overall and by age ranges, despite having an infection prevalence below the national average–albeit the SCFR-Mo does not reach statistical significance. The data from the Informes de Envejecimiento en Red (Networking Ageing Reports) [51] indicate that the ratio of places in nursing homes per 100 inhabitants in this CC.AA is higher than the national average (4.8 vs 4.1). In contrast, the CFR-Mo is lower than the CFR-PCR+ in all age ranges, which can mean, as compared with other C.AA, that a larger percentage of deaths are in-hospital rather than in closed institutions. It could be suggested in this case that the increased case fatality rate in this CC.AA might be related mainly to the characteristics and saturation of the hospital system. A different situation can be the CC.AA of Asturias, which shows a case fatality rate similar to the Spanish average in people under age 65, and the second and third highest case fatality rate in age 65–74 and over age 74, respectively. The average age of the population of Asturias is higher than Spain's, and with a greater prevalence of comorbidities [12]. Taking into account that the CFR-Mo is above the CFR-PCR+, it is likely that a dispersed population, a poorer accessibility due to orographic features, and comorbidities have been the causes of the high case fatality rate in older population, especially those institutionalized. In people older than 74, the difference between the CFR-Mo and the CFR-PCR + has been specially evident in Castilla La Mancha, Castilla y León, Asturias, Cataluña, and Madrid. In these first two CC.AA, the ratio of places in nursing homes doubles the national average [50], whereas in the last two C.AA, the increased case fatality rate in those over 74 years of age has its origin in the saturation of the hospital system and in the large absolute number of institutionalized persons.

This study has some limitations. The coordination between the different C.A and the central government can affect the report of deaths in the form of delays in some CC.AA. In order to reduce this bias, we have collected information considering the week of 11 to 16 May instead of a single date. On the other hand, the calculation of the case fatality rate with both methods

has included household deaths and deaths in nursing homes. The case fatality rate in institutions presents a wide variability between the different studies, and additionally, they do not use a common methodology. Nevertheless, this strategy better reflects the pandemic and does not represent a limitation for the comparison between CC.A. That every person who dies with a positive diagnostic result for SARS-CoV-2 is considered a death attributable to covid19 implies some classification bias. Additionally, when calculating the MoMo case fatality rate, we assume that the total excess mortality is attributable to the SARS- CoV-2 infection. There are no available data yet that allow determining which cases of the excess mortality are a direct cause of COVID-19, which are an indirect effect, and which are independent. However, it is likely that this distribution of the excess mortality has not differed across the different CC.AA. potential differences among Regions cannot be excluded. Although some potential differences among Regions in protective variables like Vitamin D status, BCG vaccination, prevalence of latent tuberculosis infection can not excluded, we believe that since there is a national program, these differences do not significantly change the objectives of the study concerning comparisons between Regions.

In conclusion, the estimated impact of the pandemic using only the case fatality rate by PCR+ cases underestimates its severity, so a more appropriate method for comparing the case fatality rate between CC.AA is the MoMo system. The population older than 75 present a case fatality rate 130 times higher than the population between 20 and 65 years of age. As a result, the protection of older population, especially those with cardiovascular or respiratory comorbidity, or diabetes, and those institutionalized in nursing homes, should be a priority in the healthcare strategy. The key feature of the first wave in Spain was the comprehensive lockdown for all the population, without differences between Regions. This is an excellent opportunity for another study to compare the effect of a single strategy as opposed to different strategies, as happened later on in Spain during the subsequent waves.

## Supporting information

**S1 File.**
(XLSX)

**S2 File.**
(XLSX)

## Acknowledgments

To the National Assembly of the Spanish Society of Primary Care Physicians and to the COVID-19 Group of the Spanish Society of Primary Care Physicians (SEMERGEN) (group coordinator Martín Sánchez, Vicente; vicente.martin@unileon.es, associated members Barquilla García Alfonso, Olmo Quintana Vicente, Serrano Cumplido Adalberto, Ruiz García Antonio, Morán Álvaro.

## Author Contributions

**Conceptualization:** Martín-Sánchez V., Calderón-Montero A., Barquilla-García A., Vitelli-Storelli F., Segura-Fragoso A., Olmo-Quintana V.

**Formal analysis:** Martín-Sánchez V., Vitelli-Storelli F., Segura-Fragoso A.

**Investigation:** Calderón-Montero A.

**Methodology:** Martín-Sánchez V., Calderón-Montero A., Barquilla-García A., Vitelli-Storelli F., Segura-Fragoso A., Olmo-Quintana V., Serrano-Cumplido A.

**Supervision:** Martín-Sánchez V., Barquilla-García A., Olmo-Quintana V., Serrano-Cumplido A.

**Validation:** Martín-Sánchez V., Calderón-Montero A., Barquilla-García A., Vitelli-Storelli F., Segura-Fragoso A., Olmo-Quintana V., Serrano-Cumplido A.

**Writing – original draft:** Martín-Sánchez V., Calderón-Montero A.

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
