## [Decision Letter · Decision Letter 0]

14 Jun 2021

PONE-D-21-12240

Analysis of case fatality rate of SARS-CoV-2 infection in the Spanish Autonomous Communities between March and May 2020.

PLOS ONE

Dear Dr. Calderón,

Thank you for submitting your manuscript to PLOS ONE. The reviews have been positive. After careful consideration, we feel that it has merit but does not fully meet PLOS ONE’s publication criteria as it currently stands. Therefore, we invite you to submit a revised version of the manuscript that addresses the points raised during the review process.

The reviewers comments are appended below the mail. Reviewers have generally approved the quality of work with certain reservations.  

Special attention is required to Improve the statistical analysis method section. It needs to made sufficiently detailed to make it easily understandable and reproducible by other researchers in the field (see reviewer comments).

Additionally,

1) Clarify whether for CFR calculation, the number of case and death being taken are from same day or different days (it appears to be from same week, which will only minimize the effect of reporting bias/delay, not other inherent issues, see latter). The day from symptoms appearance to outcome (resolution or mortality) takes upwards of 10 days. So for comparative analysis as presented in the paper, where the CFR of different communities are being compared at a same time point could erroneously make conclusions non dependable for any purpose if done during the wave of infections unless incubation period approximation is taken into consideration (e.g., Cases at day X, then deaths on day X+14 day, if 14 day average time is expected for the disease outcome). The non dependability could stem from different communities/regions may be at different phases of the pandemic for a variety of reasons, differential stringency of measures in place, people behavior, season etc. Therefore, author should make the description as clear as possible and include the incubation period consideration in the CFR calculation or alternatively present analysis of the data when all infections had an outcome (i.e., post wave of infections).

2)  Include background information about different protective variables ("more background information related to the determinates of outcome") being discussed in the literature [e.g., Global Health Security index (Health care access/ setup/ capacity, prevention practices etc.), COVID-19 stringency index (stringency of measures in place to prevent infections in different regions during the study period), Vitamin D status, BCG vaccination, prevalence of Latent Tuberculosis infection (LTBI)/TST positivity, etc.] that may have varied in these regions. 

3)Briefly, discuss how the protective variables (indicated in #2) may be impacting the observation/conclusions made in the current study. Wherever possible, make an attempt to correlate the  study observations with supposed protective variables.  

4) Enhance referencing for facts which are not commonly known (provide references for the global audience as specifics of the study area are not known widely).

We look forward to receiving your revised manuscript.

Kind regards,

Samer Singh, Ph.D.

Academic Editor

PLOS ONE

Journal Requirements:

2. Please include your tables as part of your main manuscript and remove the individual files. Please note that supplementary tables should be uploaded as separate "supporting information" files.

3. In your ethics statement in the Methods section and in the online submission form, please provide additional information about the data used in your retrospective study. Specifically, please ensure that you have discussed whether all data were fully anonymized before you accessed them and/or whether the IRB or ethics committee waived the requirement for informed consent. If patients provided informed written consent to have data from their medical records used in research, please include this information.

5. One of the noted authors is a group or consortium [COVID-19 Group of the Spanish Society of Primary Care Physicians (SEMERGEN)]. In addition to naming the author group, please list the individual authors and affiliations within this group in the acknowledgments section of your manuscript. Please also indicate clearly a lead author for this group along with a contact email address.

Reviewers' comments:

Reviewer's Responses to Questions

**Comments to the Author**

1. Is the manuscript technically sound, and do the data support the conclusions?

Reviewer #1: Yes

Reviewer #2: Yes

Reviewer #3: Partly

Reviewer #4: Yes

2. Has the statistical analysis been performed appropriately and rigorously? 

Reviewer #1: Yes

Reviewer #2: Yes

Reviewer #3: Yes

Reviewer #4: Yes

3. Have the authors made all data underlying the findings in their manuscript fully available?

Reviewer #1: Yes

Reviewer #2: Yes

Reviewer #3: Yes

Reviewer #4: Yes

4. Is the manuscript presented in an intelligible fashion and written in standard English?

Reviewer #1: Yes

Reviewer #2: Yes

Reviewer #3: Yes

Reviewer #4: Yes

5. Review Comments to the Author

Reviewer #1: 

Overall Comments:

This is a very interesting study, and the statistical inputs are adequately discussed. There are only a couple of minor statistical concerns about this paper.

Statistical critiques:

1. The statistical methods used in the paper were adequately presented; however, there is no Statistical Analysis Section in this paper. In the current version, all the statistical methods, such as Spearman’s Rho Coefficient, linear regression, etc., were spread out in the manuscript. This paper would have been strengthened by centralizing all the statistical inputs in one section – Statistical Analysis Section.

2. Please clarify the method that was used to generate the 95% confidence intervals in Figures 1 - 3 and include this information in the Statistical Analysis Section.

Reviewer #2: I would like to mention the following comments:

1-Abstract: ENE and R2 need to be defined.

2- Introduction: The second paragraph, the first sentence needs reference.

3- Method: The reliability of data is not known.

4- Method: Statistical analysis and the applied tests are missing.

5-Figure 1: MoMo??

6- The names of tables must be at the top of tables.

Reviewer #3: Reviewer's report

Overall comments:

This was an excellent report on very methodical research. The literature review was comprehensive, the methodology was painstakingly thorough and incorporated the use of sufficient numbers of samples in estimating case fatality rate of SARS-CoV-2 infection different method of the official records and the daily mortality record. However, I have concerns that the authors did not completely address in their analysis. In particular, I’m interested in clarifying the research question in specific point that addresses discrepancy in reporting CFR across different methods or correlates the prevalence of infection to CFR.

Moreover, we would like to come across more background information related to the determinates of outcome across the Spanish Autonomous Communities (CC.AA (health services and interventions).

Research question:

• In your abstract, you stated that the objective of of this paper is to compare COVID-19 case fatality rates across the Spanish CC.AA. It obvious to find discrepancy for the same outcome across different communities characterized by different socio-economic and demographic features. However, I think the paper aims do detect differences in CFR for Covid-19 in the Spanish Autonomous Communities (CC.AA) when using different methods of the official records and the daily mortality record. Moreover, if your primary outcome is to estimate the CFR, then the prevalence of infection for COVID-19 will be treated as a secondary outcome that necessary to estimate CFR.

- Please clarify your study question, whether to correlate the estimated prevalence of infection with CFR or to compare CFR across different methods of CFR-PCR+ and CFR-Mo.

Introduction:

• Please mention any variation in preventive measures across the State Administration and the Autonomous Communities (CC.AA.) in your background information; such as health services, social distancing, closure of public transport, workplaces, and schools, and termination of public gatherings and events.

• You can move some background information about the official records and the daily mortality record from introduction to methods and material section.

Material and Methods:

• Although the number of deaths for the calculation of CFR-PCR+ was obtained from the information provided by the Ministry of Health during the week of 11 to 16 May 2020, the period for data collection on the infection prevalence was not identified. Due to the rapid spreads of pandemic, inconsistency between the period of collecting the data on infection prevalence and the period of estimating CFR may be affect the study results.

-Please identify the period for the data collection on the prevalence of positive IgG antibody tests of the ENE-COVID-19 to estimate the infection prevalence.

• You concluded, “It is preferable to consider the daily mortality records to estimate case fatality rate because the official records underestimates the case fatality rate of the SARS-CoV-2 virus pandemic”. However, more information about the validly and process of collecting the data in both methods is required.

-Please clarify any auditory methods or measures taken to classify deaths in CFR data, such as ICD-10, with clear exclusion and inclusion criteria and specific identification for the main cause and underlying cause of death.

Reviewer #4: It would be appropriate if few of the sentences are rephrased and checked for grammatical errors.

6. PLOS authors have the option to publish the peer review history of their article (what does this mean?). If published, this will include your full peer review and any attached files.

Reviewer #1: No

Reviewer #2: **Yes**

Reviewer #3: **Yes**

Reviewer #4: No

---

## [Author Response · Author response to Decision Letter 0]

29 Oct 2021

Question Answer

Clarify whether for CFR calculation, the number of case and death being taken are from same day or different days (it appears to be from same week, which will only minimize the effect of reporting bias/delay, not other inherent issues, see latter). The day from symptoms appearance to outcome (resolution or mortality) takes upwards of 10 days. So for comparative analysis as presented in the paper, where the CFR of different communities are being compared at a same time point could erroneously make conclusions non dependable for any purpose if done during the wave of infections unless incubation period approximation is taken into consideration (e.g., Cases at day X, then deaths on day X+14 day, if 14 day average time is expected for the disease outcome). The non dependability could stem from different communities/regions may be at different phases of the pandemic for a variety of reasons, differential stringency of measures in place, people behavior, season etc. Therefore, author should make the description as clear as possible and include the incubation period consideration in the CFR calculation or alternatively present analysis of the data when all infections had an outcome (i.e., post wave of infections). 

The data collected come from the Spanish Ministry of Health and are harmonized for all Spanish Regions. Since these data are retrospective, we understand that they are not affected by the clinical course, including the incubation period. In this case, the effect would be the same on all Regions, given that during the first wave, all the Regions followed the guidelines of the Spanish central government and did not have independence of actions. This aspect is highlighted in Material and Methods section, where we also point out that the data are collected within a window of one week, in order to minimize the possible bias in the report of cases and deaths by the Regions.

) Include background information about different protective variables ("more background information related to the determinates of outcome") being discussed in the literature [e.g., Global Health Security index (Health care access/ setup/ capacity, prevention practices etc.), COVID-19 stringency index (stringency of measures in place to prevent infections in different regions during the study period), Vitamin D status, BCG vaccination, prevalence of Latent Tuberculosis infection (LTBI)/TST positivity, etc.] that may have varied in these regions. 

The general guidelines of the Spanish Health System (SNS) regarding vaccination, prevention protocols, and public health measures are agreed by the Interterritorial Board of the SNS, which includes the Ministry of Health and the Regional Health Departments. Therefore, as regards vaccination plans (such as BCG) or the prevalence of tuberculosis infection, although potential differences among Regions cannot be excluded, we believe that since there is a national program, these differences do not significantly change the objectives of the study concerning comparisons between Regions. As for other protective variables such as Vitamin D, there is not enough scientific information from the pandemic period to analyze the differences between Regions. We include the corresponding annotations in the manuscript.

Briefly, discuss how the protective variables (indicated in #2) may be impacting the observation/conclusions made in the current study. Wherever possible, make an attempt to correlate the study observations with supposed protective variables. 

The answer is related to the previous point. In any case, we include this limitation to the manuscript.

Enhance referencing for facts which are not commonly known (provide references for the global audience as specifics of the study area are not known widely). 

The main objective of the study is to analyze if there are differences in case fatality rates, by comparing the “official” register of the Ministry of Health and the MoMo mortality register, as well as to compare if there are differences between Regions between both register systems. From the data obtained, a number of options are considered as possible causes of the differences observed. In the ‘Discussion’ section, we consider such factors as population dispersion, the distribution and ageing of population in the different Regions, and the accessibility to hospital care and emergency care services, among others. To what extent these and other factors are responsible for the differences found should be addressed by another study. 

The statistical methods used in the paper were adequately presented; however, there is no Statistical Analysis Section in this paper. In the current version, all the statistical methods, such as Spearman’s Rho Coefficient, linear regression, etc., were spread out in the manuscript. This paper would have been strengthened by centralizing all the statistical inputs in one section – Statistical Analysis Section. 

We agree with the reviewer’s suggestion and make the corresponding correction.

Abstract: ENE and R2 need to be defined 

We make the changes suggested by the reviewer.

Introduction: The second paragraph, the first sentence needs reference 

The sentence’s reference is a previous reference from Lancet 2020: (2).

Method: The reliability of data is not known. 

In the ‘Material and Methods’ section, we refer to the reliability of data, which come from the official publications of the Ministry of Health and the Regional Health Departments. Analyzed from a current perspective, the data used have been verified and confirmed by the official national and regional bodies.

5-Figure 1: MoMo?? 

We made the correction

In your abstract, you stated that the objective of of this paper is to compare COVID-19 case fatality rates across the Spanish CC.AA. It obvious to find discrepancy for the same outcome across different communities characterized by different socio-economic and demographic features. However, I think the paper aims do detect differences in CFR for Covid-19 in the Spanish Autonomous Communities (CC.AA) when using different methods of the official records and the daily mortality record. Moreover, if your primary outcome is to estimate the CFR, then the prevalence of infection for COVID-19 will be treated as a secondary outcome that necessary to estimate CFR.

 - Please clarify your study question, whether to correlate the estimated prevalence of infection with CFR or to compare CFR across different methods of CFR-PCR+ and CFR-Mo. 

We agree with the reviewer that the objective of the study may not be clearly defined in the ‘Abstract’. According with your accurate comment, the main objective is to detect whether there are any differences in case fatality rates between Spanish Regions using two different register systems, i. e., the official register of the Ministry of Health and the MoMo; and secondarily, to analyze whether the prevalence of infection can explain those differences. We make the necessary corrections.

Introduction:

 • Please mention any variation in preventive measures across the State Administration and the Autonomous Communities (CC.AA.) in your background information; such as health services, social distancing, closure of public transport, workplaces, and schools, and termination of public gatherings and events. 

The key feature of the first wave in Spain was the comprehensive lockdown for all the population, without differences between Regions. This is an excellent opportunity for another study to compare the effect of a single strategy as opposed to different strategies, as happened later on in Spain during the subsequent waves.

You can move some background information about the official records and the daily mortality record from introduction to methods and material section. 

The data collection system of the Ministry of Health and of the mortality register is specified in the ‘Materials and Methods’ section. During the pandemic, there was a logical oversaturation of information, which however, looking back, did not affect its accuracy and reliability.

Material and Methods:

• Although the number of deaths for the calculation of CFR-PCR+ was obtained from the information provided by the Ministry of Health during the week of 11 to 16 May 2020, the period for data collection on the infection prevalence was not identified. Due to the rapid spreads of pandemic, inconsistency between the period of collecting the data on infection revalence and the period of estimating CFR may be affect the study results.

 The date of collection of data on cases and deaths was chosen on purpose to make it coincide with the completion of ENE-Covid19, and therefore, with the highest infection incidence known at that time. Although there may be some discrepancy, given that the data are so consecutive, we believe that this does not affect the data analysis. The dates of data collection are specified in the ‘Materials and Methods’ section, which is considered as a possible limitation in the corresponding section.

-Please identify the period for the data collection on the prevalence of positive IgG antibody tests of the ENE-COVID-19 to estimate the infection prevalence. 

The dates of data collection of ENE-Covid19 are clarified in ‘Materials and Methods’.

You concluded, “It is preferable to consider the daily mortality records to estimate case fatality rate because the official records underestimates the case fatality rate of the SARS-CoV-2 virus pandemic”. However, more information about the validly and process of collecting the data in both methods is required. 

This dates are clarified in ‘Materials and Methods’

Please clarify any auditory methods or measures taken to classify deaths in CFR data, such as ICD-10, with clear exclusion and inclusion criteria and specific identification for the main cause and underlying cause of death. 

In the Spanish register system, every person who dies with a positive diagnostic result for SARS-CoV-2 is considered a death attributable to covid19, and that is how it has been considered for the analysis. Obviously, this implies some classification bias, which is commented on in the ‘Discussion’ and in a previous article by the same group of authors.

Reviewer #4: It would be appropriate if few of the sentences are rephrased and checked for grammatical errors. 

We review the paper in order to make the necessary corrections according to the reviewer’s comment.

---

## [Decision Letter · Decision Letter 1]

17 Nov 2021

Analysis of case fatality rate of SARS-CoV-2 infection in the Spanish Autonomous Communities  between March and May 2020.

PONE-D-21-12240R1

Dear Dr. Calderón,

We’re pleased to inform you that your manuscript has been judged scientifically suitable for publication and will be formally accepted for publication once it meets all outstanding technical requirements.

Kind regards,

Samer Singh, Ph.D.

Academic Editor

PLOS ONE

Additional Editor Comments (optional):

Reviewers' comments:

Reviewer's Responses to Questions

**Comments to the Author**

1. If the authors have adequately addressed your comments raised in a previous round of review and you feel that this manuscript is now acceptable for publication, you may indicate that here to bypass the “Comments to the Author” section, enter your conflict of interest statement in the “Confidential to Editor” section, and submit your "Accept" recommendation.

Reviewer #1: All comments have been addressed

Reviewer #2: All comments have been addressed

Reviewer #3: All comments have been addressed

Reviewer #4: All comments have been addressed

2. Is the manuscript technically sound, and do the data support the conclusions?

Reviewer #1: Yes

Reviewer #2: Yes

Reviewer #3: Yes

Reviewer #4: Yes

3. Has the statistical analysis been performed appropriately and rigorously? 

Reviewer #1: Yes

Reviewer #2: Yes

Reviewer #3: Yes

Reviewer #4: Yes

4. Have the authors made all data underlying the findings in their manuscript fully available?

Reviewer #1: Yes

Reviewer #2: Yes

Reviewer #3: Yes

Reviewer #4: Yes

5. Is the manuscript presented in an intelligible fashion and written in standard English?

Reviewer #1: Yes

Reviewer #2: Yes

Reviewer #3: Yes

Reviewer #4: Yes

6. Review Comments to the Author

Reviewer #1: The authors have responded well to the statistical issues raised in the previous review. There is no further statistical concern about this revised manuscript.

Reviewer #2: Considering the answers of authors to my comments, I agree that the authors have been addressed all my comments.

Reviewer #3: I do acknowledge the point-by-point response made by authors to address reviewer’s comments and I am satisfied with the improvements.

No further comments.

Reviewer #4: (No Response)

7. PLOS authors have the option to publish the peer review history of their article (what does this mean?). If published, this will include your full peer review and any attached files.

Reviewer #1: No

Reviewer #2: **Yes: **Masoud Amiri

Reviewer #3: **Yes: **Adam IF

Reviewer #4: No

---

## [Editor Report · Acceptance letter]

26 Nov 2021

PONE-D-21-12240R1 

Analysis of case fatality rate of SARS-CoV-2 infection in the Spanish Autonomous Communities between March and May 2020. 

Dear Dr. A:

I'm pleased to inform you that your manuscript has been deemed suitable for publication in PLOS ONE. Congratulations! Your manuscript is now with our production department. 

Kind regards, 

on behalf of

Dr Samer Singh 

Academic Editor

PLOS ONE